# Unique ecology of co-occurring functionally and phylogenetically undescribed species in the infant oral microbiome

Nicholas Pucci[1], Amke Marije Kaan [2], Joanne Ujčič-Voortman[3], Arnoud P. Verhoeff[3,4], Egija Zaura[2], Daniel R. Mende [1,5]*

1 Department of Medical Microbiology & Infectious Diseases, Amsterdam UMC, Location AMC, Amsterdam, The Netherlands, 2 Department of Preventive Dentistry, Academic Center for Dentistry Amsterdam, University of Amsterdam and Vrije Universiteit Amsterdam, Amsterdam, The Netherlands, 3 Sarphati Amsterdam, Public Health Service Amsterdam, Amsterdam, The Netherlands, 4 Department of Sociology, University of Amsterdam, Amsterdam, The Netherlands, 5 Human Biology-Microbiome-Quantum Research Center (WPI-Bio2Q), Keio University, Tokyo, Japan

* d.r.mende@amsterdamumc.nl, mende@keio.jp

## Abstract

Early-life oral microbiome development is a complex community assembly process that influences long-term health outcomes. Nevertheless, microbial functions and interactions driving these ecological processes remain poorly understood. In this study, we analyze oral microbiomes from a longitudinal cohort of 24 mother-infant dyads at 1 and 6 months postpartum using shotgun metagenomics. We identify two previously undescribed *Streptococcus* and *Rothia* species to be among the most prevalent, abundant and strongly co-occurring members of the oral microbiome of six-month-old infants. By leveraging metagenome-assembled genomes (MAGs) and genome-scale metabolic models (GEMS) we reveal their genomic and functional characteristics relative to other infant-associated species and predict their metabolic interactions within a network of co-occurring oral taxa. Our findings highlight unique functional features, including genes encoding adhesins and carbohydrate-active enzymes (CAZymes). Metabolic modeling identified potential exchange of key amino acids, particularly ornithine and lysine, between these species, suggesting metabolic cross-feeding interactions that may explain their co-abundance across infant oral microbiomes. Overall, this study provides key insights into the functional adaptations and microbial interactions shaping early colonization in the oral cavity, providing testable hypotheses for future experimental validation.

## Author summary

We investigated how infant oral bacterial communities develop during their first six months of life, with the aim to understand which microbes colonize, how they establish themselves and why they succeed together. Using high throughput

**Data availability statement:** The raw-reads for each AIMS biosample were submitted to the European Nucleotide Archive (ENA) at EMBL-EBI repository and the National Center for Biotechnology and Information (NCBI with BioProject accession number PRJEB88622). The code generating the main figures and underlying data are available at GitHub: https://github.com/nicholaspucci/infant-oral-microbiome-strep-rothia.

**Funding:** This project was supported by the City of Amsterdam to APV, the University of Amsterdam - Research Priority Area Personal Microbiome Health (RPA-PMH) to EZ, the Stichting Orale Biologie to NP and the Dutch Research Council (NWO) - MetaHealth project (NWA.1389.20.080) as well as the Japan Society for the Promotion of Science(JSPS) - Human Biology-Microbiome-Quantum Research Center (WPI-Bio2Q). The funders had no role in study design, data collection and analysis, decision to publish, or preparation of the manuscript.

**Competing interests:** The authors have declared that no competing interests exist.

DNA sequencing techniques, we analyzed oral samples from 24 mother-infant pairs at one and six months after birth. We found two abundant, but previously unknown bacterial species (one Streptococcus spp. and one Rothia spp.) at six months of age. These bacteria consistently appear together across different babies, suggesting they may depend on each other for survival and growth. By reconstructing the genomes of these bacteria directly from our samples, we discovered specific genetic features that help explain their success in the infant mouth. *Streptococcus* carries genes involved in amino acid biosynthesis (including arginine biosynthesis using amino acids present in breast milk) as well as enzymes that help break down carbohydrates in the oral biofilm. *Rothia* has genes associated with cell membrane biosynthesis and carbohydrate metabolism, while producing nutrients that *Streptococcus* needs. We predict these bacteria exchange key nutrients like ornithine and lysine, creating a mutually beneficial partnership. The bacteria seem to cooperate and are predicted to exchange key nutrients like malate and lysine which may help maintain a healthy oral environment by regulating acidity levels, potentially protecting against tooth decay.

## Introduction

The first bacteria ('animalcules') Antonie van Leeuwenhoek observed in the 17th century were oral microbes. In centuries afterwards, oral microbes were found to be involved in a range of conditions, such as dental caries, periodontal diseases and oral candidosis [1]. An understanding of the beneficial role that the microbes constituting the oral microbiome play was realized much later. A balanced oral microbiome promotes oral health by serving as a protective barrier, preventing infections and colonization of oral surfaces by harmful pathogens as well as regulating oral pH [2, 3]. Beyond oral health, the oral microbiome has significant implications for systemic health [4], contributing to vasodilation and blood pressure regulation through nitrogen and enterosalivary nitrate metabolism [5]. Though its importance is undeniable, our knowledge about the oral microbiome, particularly its development during infancy, remains limited, and public metagenomics datasets remain scarce.

Most of the existing knowledge about the infant oral microbiome stems from 16S rRNA gene amplicon surveys, which revealed the genus composition of the oral microbiome development through a predictable succession of microbial colonization, where early colonizers such as *Streptococcus* and *Actinomyces* adhere to oral tissues thanks to their adhesin repertoire [6–11]. These pioneers subsequently form a foundation that supports the establishment of secondary colonizers, including *Veillonella* and *Rothia* [12]. Developmental milestones, such as primary dentition and environmental exposures are often accompanied by further compositional shifts [8]. While these coarse grained taxonomic succession patterns have been described, the functional potential within these microbial communities, particularly during the formative months of life, remain poorly characterized. Yet, such knowledge about the

developing oral microbiome would be crucial given the potential persisting effects on oral and systemic health later in life [8, 13, 14].

Beyond individual species functions, microbial interactions fundamentally shape microbiome structure and function [12, 15], yet little is known about their contribution to the stability and resilience of the developing oral microbiome. Typically, these interactions are a combination of spatial organization [16, 17], complex metabolic dependencies and cross-feeding mechanisms [18] leading to mutualistic, competitive or commensal relationships [3, 19–21] that drive community-level processes such as pH regulation [22] and nitrate reduction [23]. Species that consistently co-occur across individuals may share underlying ecological dependencies, such as metabolic cross-feeding or niche overlap [24, 25]. Understanding the genomic basis of these synergistic activities, including metabolic pathways and genetic determinants, is crucial for deciphering how microbial communities establish and maintain functional stability during infant oral microbiome development.

Methodological advances allowing for the reconstruction of genomes directly from metagenomics, and their interrogation through comparative approaches (metapangenomics) [26] have the potential to provide insight into the functional potential and metabolic interactions of the developing microbiome. For example, genome-resolved metagenomic approaches allowed Utter et al. [26] to have strain-level resolution and identification of accessory genes conferring adaptive advantages to *Rothia* and *Haemophilus* spp. to specific oral niches in healthy adults. Further, novel tools allow for the generation of accurate metabolic models directly from genomes [27, 28], which allows the prediction of substrate utilization and metabolite production. Community models combining the metabolic models of multiple co-occurring microbes can provide further insight into metabolic interactions and cross-feeding, therefore elucidating the metabolic basis for the synergistic activities and successional patterns observed during oral microbiome development [29–31].

Here, we performed shotgun metagenomics on a time series of 24 infants and their mother's oral microbiome and identified novel, co-occurring species using a meta-pangenomics approach. Analysis of a time-series of tongue dorsum and dental biofilm samples collected from children up to six months of age and their mothers as part of the prospective Amsterdam Infant Microbiome Study (AIMS) revealed previously undescribed *Streptococcus* and *Rothia* species to be predominant members of the infant oral cavity. By leveraging metagenome-assembled genomes (MAGs), we characterize genomic features and functional attributes of these species to explain their adaptations to the infant oral cavity. In order to interpret their co-abundance patterns we further decipher their metabolic interactions within an oral species network. Overall, our findings highlight important microbial ecological dynamics in the developing oral ecosystem of infants and establish a foundation for understanding how early microbial interactions help shape the infant oral microbiome.

## Results

### Undescribed *Streptococcus* and *Rothia* species emerge as abundant members of the six-month infant oral microbiome

We investigated the oral microbiome of 24 mother-infant pairs from the AIMS cohort, focusing on full-term, vaginally-born infants with no reported antibiotic exposure (S1 Table). By six months, 87% of AIMS infants (21 out of 24) had obtained their first teeth, allowing collection of dental biofilm. Shotgun metagenomics sequencing (mean depth tongue biofilm: 20.5M reads per sample; mean depth dental biofilm: 15.7M reads per sample; S1 Fig) was performed on samples collected from tongue dorsum swabs from mothers (successfully sequenced samples: n = 22) and infants both 1 (n = 9) and 6 months (n = 18), and dental biofilm from mothers (n = 21) and infants at 6 months (n = 14). All analyzed one-month-old infants were exclusively breastfed. By six months, we observed diverse feeding patterns including breast- (n = 8), mixed-(n = 7) and formula feeding (n = 5), with 95% (21 of 22) of infants being introduced to solid foods (S1 Table).

Metagenomic profiling revealed the *Streptococcus* genus to be the most abundant and prevalent taxonomic group in the oral microbiome of AIMS infants at both 1 and 6 months of age. However, we observed a notable species-level compositional shift over time, with previously undescribed species emerging by six months. At one month, infant tongue dorsum hosted an average of 19 (±6) bacterial species (Fig 1B) and was predominantly composed of *Streptococcus*

**Fig 1. Oral microbiome abundance and composition of AIMS mothers and their infants. A)** Relative abundance of the 5 most abundant bacterial genera detected in the tongue dorsum (TD) and dental plaque (DP) from mothers at 34wks gestation (TD: n = 22; DP: n = 21) and their infants at one (TD: n = 9) and six months (TD: n = 18; DP: n = 14) of age. Other low abundance genera are displayed in gray. **B)** Relative abundance (mean±SD) of the 15 most abundant bacterial species detected in the TD and DP of mothers (34wks gestation) and their infants (one and six months of age). Species displayed include all taxa that ranked among the 15 most abundant in at least one sample group or time point. Bars are coloured by genus. Species richness (mean±SD number of detected species) is displayed for each time point and sample type. **C)** Principal coordinate analysis (PCoA) based on Bray-Curtis dissimilarity showing microbiome compositional variation among sample types and timepoints. TD samples are represented as circles, whilst DP as triangles. Colors indicate timepoint and subject (mothers 34wks gestation = brown; infant 1 month = green; infant 6 months = red).

(relative abundance±standard deviation, SD 57.0 ± 6.8%), *Veillonella* (9.0 ± 2.0%), *Lactobacillus* (6.1 ± 3.3%) and *Rothia* (4.9 ± 1.8%) species (Figs 1A and S2). By six months, we observed an increase in bacterial species richness (tongue dorsum 43 ± 17, dental biofilm 43 ± 14) with an overall shift in tongue microbiome composition relative to one month (Figs 1A-1C and S3). Notably, previously undescribed *Rothia* (GTDB: sp902373285: relative abundance = 9.2 ± 2.2%, prevalence = 61%) and *Streptococcus* (GTDB: sp000187445: 5.6 ± 1.9%, 66%) species, hereon referred to as *Streptococcus* AIMSoral1 and *Rothia* AIMSoral2, emerged as abundant and prevalent bacteria in the oral cavity, particularly in the tongue biofilm (Figs 1B and S2). Dental biofilm composition at six months (despite the emergence of teeth as a new niche) largely resembled the tongue biofilm composition (Fig 1C). Milk feeding type did not significantly influence community composition (S4 Fig). Maternal dental and tongue biofilm communities were markedly different (Fig 1C) and strain sharing was found to be rare with only two events found across the whole cohort (S2 Table), both between mothers and their respective one-month-old infants.

### *Streptococcus* AIMSoral1 and *Rothia* AIMSoral2 show significant co-abundance in infant oral ecosystems

To elucidate potential species interactions, we computed a co-abundance network of species present in at least 5 samples (as indicated by a sensitivity analysis, see Methods) using SPIEC-EASI for tongue and dental biofilm samples, separately (S5-S6 Figs and S3-S4 Tables). We corroborated these associations using Spearman's rank correlations and retained only relationships found significant by both methods (S3 Table and S7 Fig). As the overlap in species presence between the infant and maternal oral microbiomes was minimal (Fig 1B), we only used infant samples to generate the co-abundance network. Surprisingly, in one-month-old edentate infants we could not detect any significant co-abundance interactions even though several species, including *L. paragasseri*, *S. salivarius*, *S. oralis* BN and *V. nakazawae* were found to be abundant and present in more than 5 samples.

By six months of age, both tongue and tooth surfaces displayed substantially more complex co-abundance networks (S5 Fig). We were particularly interested in the interaction partners of *Streptococcus* AIMSoral1 and *Rothia* AIMSoral2, revealing a strong pairwise association (SPIEC-EASI covariance coefficient (cov): 0.21; Spearman's rank correlation: Rho = 0.78, p = 0.0001; S6-S8 Figs and S4-S6 Tables) between these two uncharacterized species on the tongue dorsum. *Streptococcus* AIMSoral1 was the strongest association of *Rothia* AIMSoral2 and vice versa (S6 Fig). Further, while both species had other, weaker, associations with multiple species, they only shared one significant association partner, namely the much lower abundant *Pauljensenia* sp900541895 (S6 Fig; *Streptococcus* AIMSoral1 - *Pauljensenia* sp900541895: cov = 0.06, Rho = 0.59, p = 0.01; *Rothia* AIMSoral2 - *Pauljensenia* sp900541895: cov = 0.02, Rho = 0.59, p = 0.01; S4-S6 Tables). Beyond these positive associations, *Streptococcus* AIMSoral1 (cov = 0.08, Rho = -0.63, p = 0.005) and *Rothia* AIMSoral2 (cov = 0.08, Rho = -0.67, p = 0.001; S8 Fig and S4-S6 Tables) both exhibited negative co-abundance associations with *Campylobacter concisus*, a species associated with inflammatory bowel disease [32]. Additionally *Streptococcus* AIMSoral1 showed a negative association with *Prevotella melaninogenica* (cov = 0.16, Rho = -0.67, p = 0.002.; S8 Fig and S4-S6 Tables), a species implicated in oral inflammation and periodontitis in adults [33].

The detected three-species module, centred on the strong association between the undescribed *Streptococcus* AIMSoral1 and *Rothia* AIMSoral2, prompted us to further investigate their taxonomic identity and functional basis for co-occurrence.

### Genomic relatedness analyses reveal undescribed *Streptococcus* and *Rothia* to be novel species

To gain insight into the functional underpinnings driving the co-abundance patterns of the uncharacterized species, we reconstructed metagenome-assembled genomes (MAGs; S9 Table) from maternal and infant oral samples. Metagenomic binning yielded 20 *Streptococcus* and 51 *Rothia* medium and high-quality MAGs (>50% completeness and <10% contamination). Of these, five MAGs were taxonomically assigned to *Streptococcus* AIMSoral1 while 15 were

assigned to *Rothia* AIMSoral2 using GTDB-tk. As we could not recover any high or medium quality MAGs of *Pauljensenia* sp900541895, we focused on *Streptococcus* AIMSoral1 and *Rothia* AIMSoral2.

To verify these taxonomic assignments, we reconstructed phylogenomic trees based on single-copy core gene protein sequences (Fig 2), incorporating reference genomes and independently assembled MAGs from public databases (S7 Table) as well as the 15 additional *Streptococcus* and 36 *Rothia* MAGs reconstructed in this study and assigned to congeneric taxa (S7 Table). Hereafter, reference genomes, independently assembled MAGs and MAG reconstructed in this study will be referred to as 'genomes', unless otherwise specified.

Phylogenomic analysis of 89 *Streptococcus* species (110 genomes) placed *Streptococcus* AIMSoral1 within the *salivarius* group with high confidence (S9 Fig). To refine this placement, we employed a pangenomic approach focusing specifically on the *salivarius* group, constructing a phylogenomic tree from single-copy core genes (Fig 2A). This targeted analysis included 32 genomes (reference genomes = 24, MAGs = 8): *S. salivarius* (n = 7), *S. vestibularis* (n = 5), *S. thermophilus* (n = 5) and *Streptococcus* AIMSoral1 (MAGs = 14) and *S. mitis* (n = 1, used as outgroup). The phylogenetic analysis revealed *Streptococcus* AIMSoral1 to form a distinct, well-supported monophyletic clade, representing a sister clade to the *salivarius* group. This observation was supported by average nucleotide identity (ANI, species threshold = 95%) [34] analyses, confirming that *Streptococcus* AIMSoral1 represents a species-level entity, with 97.3% to 98.6% ANI within its cluster, but lower ANI to phylogenetically neighboring species (*S. salivarius*: 88.6-89.3%, *S. vestibularis*: 88.3-88.5%, *S. thermophilus*: 87.1-87.4%; S10A Fig). Notably, one *S. salivarius* genome (isolate PS4) clustered within *Streptococcus* AIMSoral1 (ANI > 97%; referred to as *S.* AIMSoral1 PS4 in S10A Fig), indicating misclassification.

The *Rothia* phylogenomic tree included 63 genomes (reference genomes = 13, MAGs = 50; Fig 2B) and placed the undescribed *Rothia* AIMSoral2 in a monophyletic clade, closely related to *R. mucilaginosa*_A and *R. mucilaginosa* [35] (Fig 2B). For *Rothia*, ANI analyses also supported *Rothia* AIMSoral2 as a species-level entity, showing 97.0% to 99.9% ANI within its cluster, but markedly lower ANI with phylogenetically neighboring *R. mucilaginosa*_A (90.2-91.4%) and *R. mucilaginosa* (86.9-88.5% ANI) (S10B Fig).

Taxonomic classification of the 16S rRNA gene sequence extracted from *Streptococcus* AIMSoral1, *Rothia* AIMSoral2 and genomes from closely related species did not resolve at species level, yielding only genus level information (S8 Table).

## Adhesion and carbohydrate utilization differentiate *Streptococcus* AIMSoral1 from other infant-associated species

To understand why *Streptococcus* AIMSoral1 is adapted to the infant oral cavity at this developmental stage, we performed comparative genomic and metabolic enrichment analyses to identify functional capabilities that distinguish it from phylogenetically related infant-associated *Streptococcus* species (see Methods: Public reference genomes acquisition).

The *Streptococcus* pangenome included 57 genomes (S7 Table) from six infant-associated species: *Streptococcus* AIMSoral1 (n = 14), *S. lactarius* (n = 12), *S. salivarius* (n = 6), *S. oralis* (n = 9), *S. mitis* (n = 12) and *S. peroris* (n = 3). *Streptococcus* AIMSoral1 genomes averaged 1.9 ± 0.9Mb (average ± SD), carrying roughly 1930 ± 98 genes. Gene-cluster (GC) analysis (S11 Fig) showed largely species-driven clustering, with *Streptococcus* AIMSoral1 displaying distinct, but close genomic similarity to *S. salivarius* (S11 Fig).

Adhesion potential, assessed via lectin-encoding gene copy number, largely mirrored species identity (Fig 3). The adhesin repertoire of *Streptococcus* AIMSoral1 showed high similarity to *S. salivarius* and *S. lactarius*, including enrichment in *fimA* lipoprotein genes (present in *Streptococcus* AIMSoral1 and *S. lactarius*, but absent in other species; Fig 3 and S9 Table).

Carbohydrate-active enzymes (CAZymes) analysis revealed *Streptococcus* AIMSoral1 harbors significantly higher copy numbers of eight glycosyltransferases (GTs), including α-N-acetylglucosaminyltransferase (GT45) and starch phosphorylases (GT35). As shown in Fig 3, *S.* AIMSoral1 displays distinct CAZyme enrichment patterns, with several glycoside hydrolases (GHs), such as GH66, GH68 and GH87 families, shared with *S. salivarius* (Fig 3 and S9 Table), while

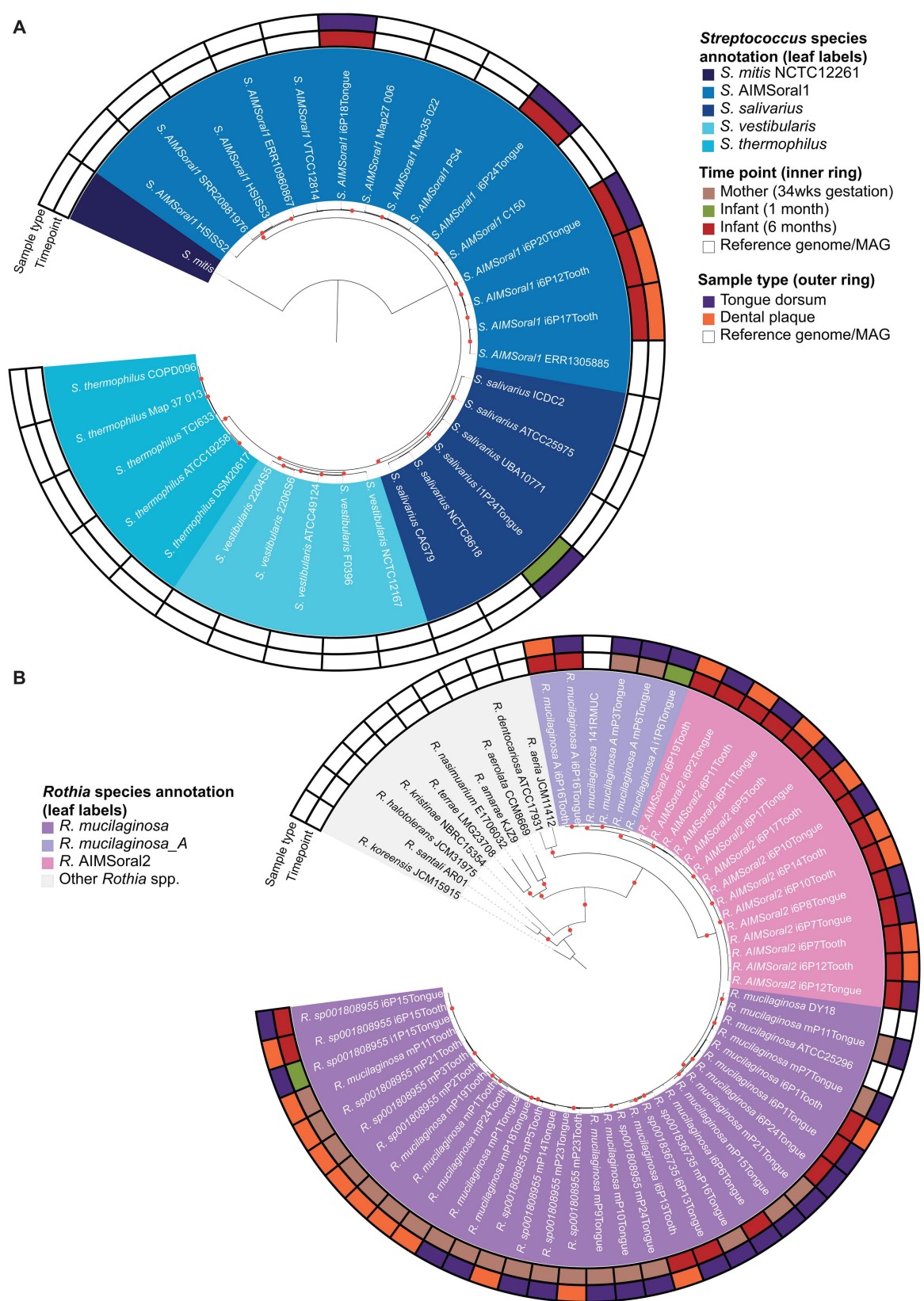

**Fig 2. Phylogenomic trees of *Streptococcus salivarius* clade and *Rothia* genus, highlighting the undescribed species *Streptococcus* AIM-Soral1 and *Rothia* AIMSoral2. A)** Phylogenomic tree of the *salivarius* clade of the *Streptococcus* genus based on concatenated amino acid sequences

of single copy-core genes from 32 genomes. The tree includes *S. salivarius* (n = 6, dark blue), *S. vestibularis* (n = 5, purple), *S. thermophilus* (n = 5, light blue), *S. mitis* (used as outgroup, green) and *Streptococcus* AIMSoral1 (n = 15, medium blue). Bootstrap ≥0.9, indicated by red circles at nodes. Concentric rings indicate sample origin: timepoint (innermost: mother 34wks gestation = brown, infant 1 month = green, infant 6 months = red) and sampling location (outermost: tongue dorsum = purple, dental plaque = orange). White ring sections indicate reference genomes. **B)** Phylogenomic tree of the *Rothia* genus based on concatenated amino acid sequences of single-copy core genes from 63 genomes from this study. Species represented include *R. mucilaginosa* (n = 32, purple), *R. mucilaginosa_A* (n = 6, pink), *Rothia* AIMSoral2 (n = 15, light purple/pink), and other *Rothia* species (shown in varying shades). Rings indicate the time point (mother 34wks gestation, infant one and six months) and sampling location (tongue dorsum and dental plaque). Red circles indicate bootstrap values≥0.9.

additionally encoding significantly higher copy-numbers of GH8, GH36 and GT2_Glyco_trans_2_3 genes compared to *S. salivarius*, suggesting distinct carbohydrate-utilization strategies. Overall, *Streptococcus* AIMSoral1 and *S. salivarius* have fewer genes involved in carbohydrate metabolism than other infant-associated species (Fig 3).

Metabolic enrichment analysis showed *Streptococcus* AIMSoral1 and *S. salivarius* to be significantly enriched in genes for amino acid metabolism compared to other infant-associated species (Figs 3 and S12). Notably, only *Streptococcus* AIMSoral1 and *S. salivarius* encoded the full arginine biosynthesis pathway from glutamate (M00028, M00844; S10 Table and S12 Fig), a downstream metabolite of the citrate cycle (also enriched; M00010; S10 Table) and major component of breast/formula milk.

### Carbohydrate utilization and fatty acid biosynthesis differentiate *Rothia* AIMSoral2 from other *R. mucilaginosa* species

We compared genomic traits of *Rothia* AIMSoral2 to infant-associated *Rothia* species to uncover metabolic adaptations supporting its predominance across 6-month-old infant oral cavities.

The *Rothia* pangenome included 53 MAGs and reference genomes (S13 Fig), from three species (as annotated by GTDB-tk), retrieved from infants in our study: *R. mucilaginosa* (30 MAGs, 2 reference genomes), *R. mucilaginosa_A* (5 MAGs, 1 reference genome) and *Rothia* AIMSoral2 (15 MAGs, no reference genomes available). *Rothia* AIMSoral2 genomes averaged 2.1±0.1Mb and carried 1760 (±37) genes. In line with phylogenomic analysis, these genomes form a consistent clade based on gene presence/absence, showing greater similarity to *R. mucilaginosa_A* than other *R. mucilaginosa* strains (S13 Fig).

All infant-associated *Rothia* species carried genes-encoding for CBM50 domains (also known as LysM; S9 Table), associated with binding bacterial peptidoglycans [36]. No matches were found in the LectomExplore database for *Rothia* lectins, suggesting limited host-surface adhesion potential. Further CAZyme analysis showed *Rothia* AIMSoral2 to be enriched in genes encoding for, among others, GT2, GT4 and GT11 [37–40](S9 Table). Additionally, *Rothia* AIMSoral2 carried significantly higher copy-numbers of redox-active enzyme-encoding genes (Auxilary Activities: AA1 and AA1_1/2; Fig 4 and S9 Table), co-acting with CAZymes.

We further investigated genes involved in the nitrate-nitrite-nitric oxide ($NO_3$-$NO_2$-NO) enterosalivary pathway in infant-associated *Rothia* species. While all studied *Rothia* species carried genes involved in the respiratory denitrification of $NO_3$ to NO and, in part, intracellular ammonia ($NH_3$) production, several pathway components (nitrate/nitrite reductases) showed modestly higher copy numbers in *R. mucilaginosa* and/or *R. mucilaginosa_A* compared to *Rothia* AIMSoral2 (Dunn's test: $p < 0.05$; Fig 4 and S9 Table). Notably, the key process of fatty acid biosynthesis initiation (M00082) was significantly enriched in *Rothia* AIMSoral2 (Fig 4 and S11 Table).

### *Streptococcus* AIMSoral1 shows higher metabolic complementarity to *Rothia* AIMSoral2 than other *Streptococcus* species

To investigate whether the co-abundance of *Streptococcus* AIMSoral1 and *Rothia* AIMSoral2 at six months of age corresponded with their metabolic interdependence, we reconstructed genome-scale metabolic models (GEMS) for

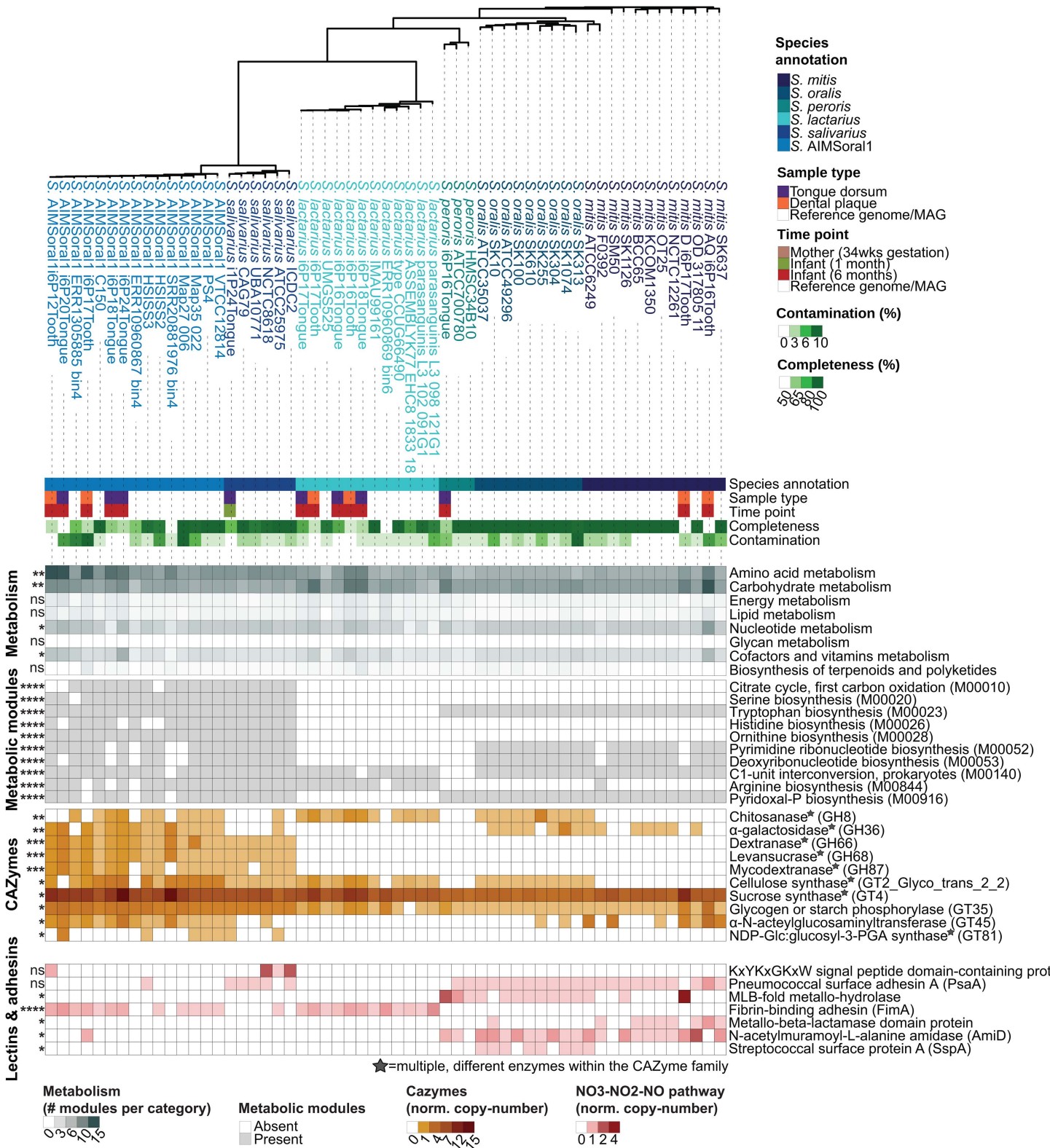

**Fig 3. Comparative genomic and functional characterization reveals unique metabolic and adhesion features of *Streptococcus* AIMSoral1 among infant-associated streptococci.** Phylogenomics tree based on single-copy core genes of 57 infant-associated *Streptococcus* genomes. Meta-data annotation bars show species annotation (color-coded), sample type (tongue=purple, dental plaque=orange), timepoint (mother=brown, infant

1mo = green, infant 6mo = red, reference = white), genome completeness (50–100%, green gradient), and contamination (0–10%, green gradient). Heatmaps below display, from top to bottom: **(1) Metabolism**: metabolic module presence across broad KEGG categories (gray scale showing normalized pathway copy-number); **(2) Metabolic modules**: presence/absence (gray/white) of specific modules, significantly enriched in *Streptococcus* AIMSoral1; **(3) CAZymes**: normalized gene copy numbers (orange scale) for carbohydrate-active enzymes. Stars denote multiple enzyme variants within families; **(4) Lectins & adhesins**: normalized gene copy numbers (red scale) for adhesion proteins detected across infant-associated streptococci. Asterisks in left margins indicate significantly higher/lower abundance in *Streptococcus* AIMSoral1 relative to other species: ns (not significant), *p < 0.05, **p < 0.01, ***p < 0.001, ****p < 0.0001 (Kruskal-Wallis with Dunn's post-hoc tests on high-quality genomes (≥90% completeness; n = 39), *fdr* correction; S9 Table).

infant-associated *Streptococcus* and *Rothia* species using the PhyloMInt tool (see Methods: Metabolic complementarity/interaction potential) and compared their metabolic complementarity ($MI_{complementarity}$) to other infant-associated congeneric species. To provide a reference point for interpreting complementarity values, we included *Phocaeiola dorei* and *Lachnoclostridium symbiosum*, a well-characterized cooperative pair from the gut microbiome [41]. This positive control analysis yielded $MI_{complementarity}$ indices of 0.10 (*P. dorei* with *L. symbiosum*) and 0.15 (*L. symbiosum* with *P. dorei*; S12 Table), providing a reference range for known metabolic cooperation against which to evaluate our findings.

Overall, the $MI_{complementarity}$ relationship between *Streptococcus* and *Rothia* species was asymmetric; *Streptococcus* species exhibited $MI_{complementarity}$ values ranging from 0.11 to 0.15 when paired with *Rothia* species, while *Rothia* species exhibited values ranging from 0.07 to 0.10 when paired with *Streptococcus* (S14A Fig and S12 Table). Notably, *Streptococcus* AIMSoral1 exhibited significantly greater $MI_{complementarity}$ with all *Rothia* species than other *Streptococcus* species (0.13 ± 0.01 to 0.15 ± 0.02; as did *S. peroris*: 0.13 ± 0.02 to 0.15 ± 0.02; S14A Fig and S12 Table). The MIcomplementarity between *Streptococcus* AIMSoral1 and *Rothia* AIMSoral2 (0.135 ± 0.017) was consistent with values from the cooperative pair *P. dorei–L. symbiosum* (0.10–0.15), suggesting a potential metabolic interaction.

Using the PhyloMInt PTM tool, we identified metabolites each species could produce and utilize (Fig 5C), allowing us to assess potential differences in metabolic interactions between *Rothia* AIMSoral2 and *Streptococcus* AIMSoral1 compared to their interactions with other infant-associated species. We predicted multiple potential metabolic interactions between *Rothia* AIMSoral2 and *Streptococcus* AIMSoral1, but these were often shared with other species (S14B Fig and Tables A and B in S13 Table).

### Genome-scale predictions of metabolic interactions in an oral bacterial co-abundance network module

Given the strong co-abundance of *Streptococcus* AIMSoral1 and *Rothia* AIMSoral2 in the six-month tongue microbiome and their shared network connections with *Pauljensenia* sp900541895, we investigated whether their co-abundance reflects metabolic interdependencies. We used GEMS to predict potential nutrient exchanges between these species, hypothesizing that metabolic interactions facilitate the co-abundance module comprising *Streptococcus* AIMSoral1, *Rothia* AIMSoral2 and *Pauljensenia* sp900541895 (Fig 5A). We inferred $MI_{complementarity}$ based on the highest quality available MAG or reference genome (*see* Methods: Prediction of metabolic complementarity and interaction potentials; S12 Table). Although we lacked sufficient *Pauljensenia* sp900541895 MAGs for comprehensive comparative genomics, metabolic prediction was still possible using an independently assembled MAG from a human gut sample [42, 43] (S7 Table). The analysis revealed asymmetric metabolic dependencies within the network (Fig 5B). *Streptococcus* AIMSoral1 exhibited $MI_{complementarity}$ values of 0.16 with *Rothia* AIMSoral2 and 0.18 with *Pauljensenia* sp900541895. In contrast, *Rothia* AIMSoral2 showed values of 0.08 with *Streptococcus* AIMSoral1 and 0.07 with *Pauljensenia* sp900541895, while *Pauljensenia* sp900541895 displayed values of 0.12 with *Streptococcus* AIMSoral1 and 0.13 with *Rothia* AIMSoral2. This asymmetric pattern, combined with predicted metabolite directionality (Fig 5C-5D), suggests that both *Rothia* AIMSoral2 and *Pauljensenia* sp900541895 may provide metabolites that support *Streptococcus* AIMSoral1 metabolism.

GEMS predicted distinct, species-level patterns of metabolite production and utilization, which are summarized in Fig 5C and Tables A and B in S13 Table. Within the co-abundance module, *Streptococcus* AIMSoral1 and *Rothia* AIMSoral2

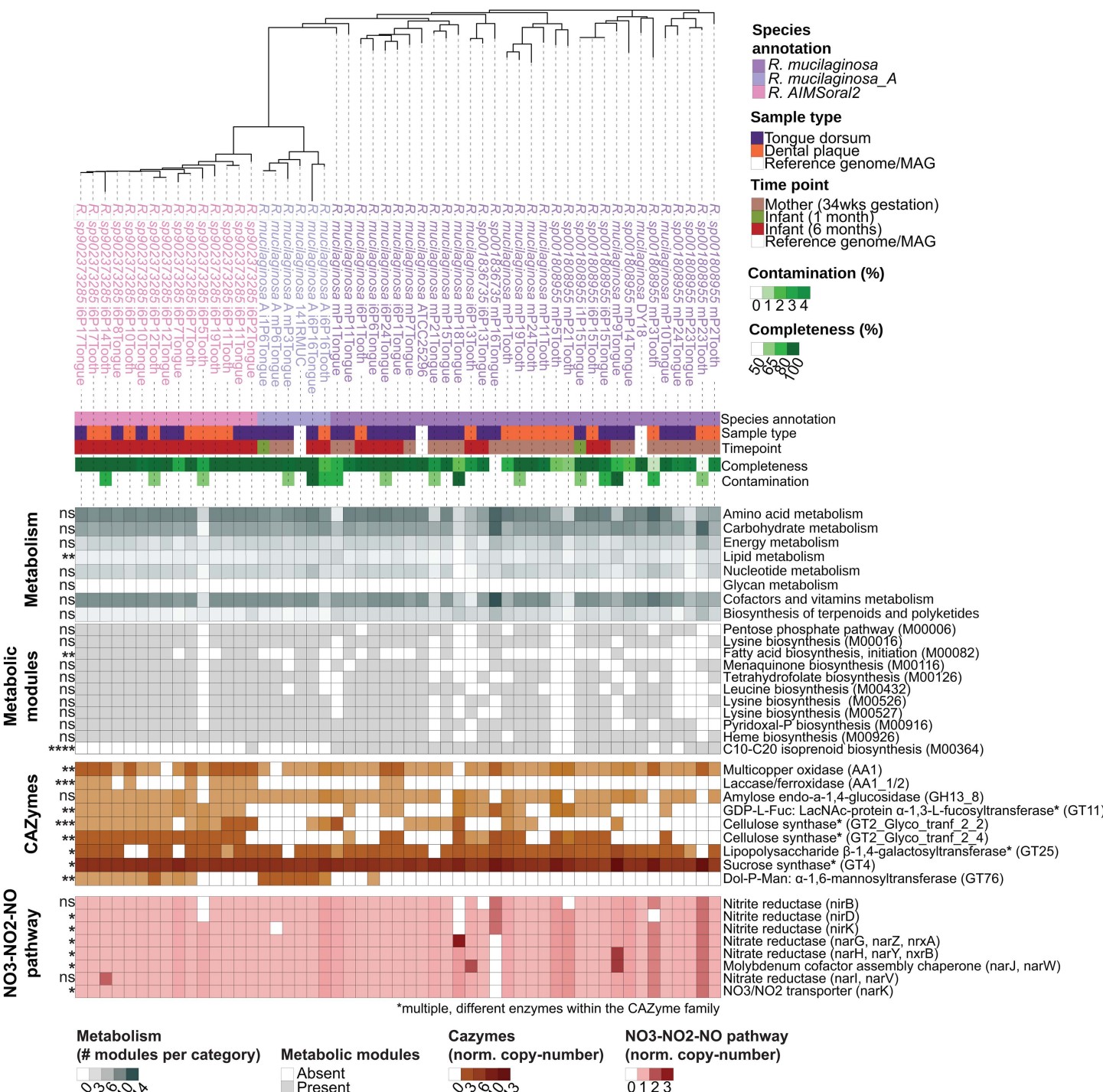

**Fig 4. Overview comparative genomics analyses outputs for *Rothia* AIMSoral2 and infant-associated *Rothia* species.** Phylogenomics tree based on single-copy core genes of 53 infant-associated *Rothia* genomes. Metadata annotation bars show species annotation (color-coded), sample type (tongue = purple, dental plaque = orange), timepoint (mother = brown, infant 1mo = green, infant 6mo = red, reference = white), genome completeness (50-100%, green gradient), and contamination (0-10%, green gradient). Heatmaps below display, from top to bottom: **(1) Metabolism**: metabolic module presence across broad KEGG categories (gray scale showing normalized pathway copy-number); **(2) Metabolic modules**: presence/absence (gray/white) of specific modules found in *Rothia* AIMSoral2 and other infant-associated Rothia spp.; **(3) CAZymes**: normalized gene copy numbers (orange scale) for carbohydrate-active enzymes. Stars denote multiple enzyme variants within families; **(4) enterosalivary nitrate (NO3-NO2-NO) pathway**:

normalized gene copy numbers (red scale) for enterosalivary pathway genes including nitrate reductase (narG, narH, narI) and nitrite reductase (nirB, nirD, nirK) detected across infant-associated Rothia species. Asterisks in left margins indicate significantly higher/lower abundance in *Rothia* AIMSoral2 relative to other species: ns (not significant), *p < 0.05, **p < 0.01, ***p < 0.001, ****p < 0.0001 (Kruskal-Wallis with Dunn's post-hoc tests on high-quality genomes (≥90% completeness; n = 40), *fdr* correction; S9 Table).

were predicted to interact through 15 metabolites (Fig 5C), of which 9 were produced by *Rothia* AIMSoral2 and utilized by *Streptococcus* AIMSoral1 and 6 produced by *Streptococcus* AIMSoral1 and utilized by *Rothia* AIMSoral2. Among these were key amino acids including nitrogen-rich lysine and ornithine. To validate predicted metabolic exchanges we systematically identified membrane transporters for metabolites showing production-utilization complementarity among the three species (Methods; S14-S15 Tables). This analysis identified encoded transporter genes supporting the predicted metabolic exchanges (Fig 5D). For key amino acids such as lysine, ornithine, serine and glutamate, we identified exporters in producing species and importers in consuming species (S15 Table), providing genomic support for directional nutrient flow. Similarly, metal ion exchanges ($Mg^{2+}$, $Mn^{2+}$, $Zn^{2+}$) were supported by the presence of appropriate uptake and efflux systems across species (Fig 5D and S15 Table).

## Discussion

In this study, we performed shotgun metagenomic sequencing on oral samples from 24 mother-infant pairs from the prospective AIMS cohort, enabling finer taxonomic resolutions than previous studies, which were mostly 16S rRNA gene amplicon-based [6, 8, 44–46]. This approach allowed the identification of previously undescribed species in the developing oral ecosystem and reconstruction of their genomes, revealing functional potential and metabolic interactions within the infant oral microbiome.

Our findings highlight the early dominance of *Streptococcus* spp. and increasing bacterial richness over time, including the emergence of the *Rothia* genus. Species-level changes in these genera, resulted in the establishment of *Streptococcus* AIMSoral1 and *Rothia* AIMSoral2 as predominant members of the infant oral microbiome. These genus-level transitions likely reflect adaptations to dietary changes during weaning and structural changes from dentition [44, 47] aligning with previous 16S rRNA gene amplicon sequencing studies [6, 8, 44, 47]. However, as both dentition and dietary transitions occur within this timeframe, more frequent longitudinal sampling would be needed to disentangle their individual contributions to the emergence of these species. Beyond these temporal considerations, our species-level observations diverge from previous findings, which suggested *R. mucilaginosa* and *S. salivarius* were highly prevalent after first dentition [6]. Conversely, six-month-old AIMS infants lacked previously reported *S. salivarius* group species and had comparatively low prevalence of *R. mucilaginosa*. This discrepancy likely stems from lower 16S rRNA resolution to distinguish *Streptococcus* AIMSoral1 and *Rothia* AIMSoral2 from *S. salivarius* and *R. mucilaginosa*, respectively (S8 Table), while our analysis identified these as separate species. We found no significant evidence that milk or formula feeding influenced the abundance of these species. To understand the ecological significance of these taxonomic shifts, we examined the functional capabilities of these predominant species, beginning with *Streptococcus* AIMSoral1.

*Streptococcus* AIMSoral1 and *S. salivarius* both showed enrichment in amino acid metabolism genes, including complete arginine and glutamate derivatives biosynthesis, suggesting adaptations to milk-derived nutrients as well as a role in oral nitrogen cycling [48–50]. However, since *S. salivarius* was notably absent from 6-month AIMS infants despite sharing these metabolic capabilities, other factors likely drive the predominance of *Streptococcus* AIMSoral1 in this niche. Previous studies have suggested the presence of the fimA adhesin (which we found encoded by *Streptococcus* AIMSoral1) to increase streptococcal specificity for the oral cavity [51, 52]. Notably, this gene was absent in *S. salivarius* and other infant-associated streptococci, except *S. lactarius*, the most abundant streptococcal species in six-month-old infants' oral cavities. *Streptococcus* AIMSoral1 also showed enrichment in several CAZymes, including dextran- and glucan-degrading

 

PLOS Computational Biology

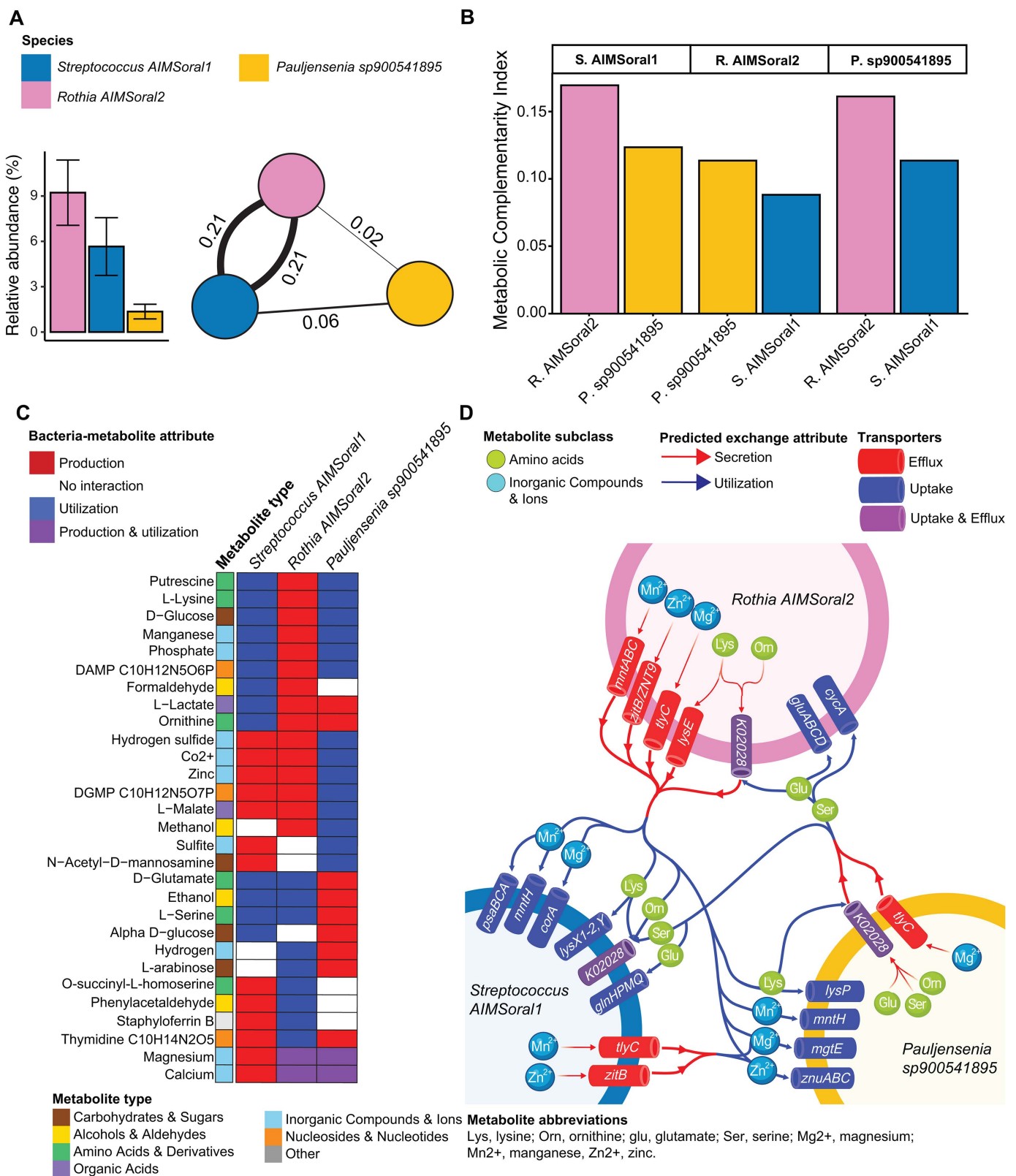

**Fig 5. Metabolic complementarity and interactions within an infant tongue microbial co-abundance network. A)** Microbial co-abundance network of three species (*Streptococcus* AIMSoral1 = blue, *Rothia* AIMSoral2 = pink, *Pauljensenia* sp900541895 = yellow), identified using SPIEC-EASI

on tongue surface samples from six-month-old infants (n = 18), with bar plot showing their mean relative abundances (±SD) at this niche. Edge width indicates strength of SPIEC-EASI covariance estimates (values shown on edges). **B)** MI$_{complementarity}$ index for pairwise interactions among species in the tongue microbial network calculated using genome-scale metabolic models (PhyloMInt). Each facet title indicates the reference species for which the index is calculated. **C)** Heatmap showing predicted bacterial-metabolite bipartite interactions (production = red, utilization = blue, both = purple, no interaction = white) potentially occurring among species in the tongue network. Metabolites are classified by compound type (amino acids and derivatives, organic acids, nucleosides and nucleotides, carbohydrates and sugar derivatives, inorganic compounds and ions, alcohols and aldehydes). **D)** Predicted metabolite exchange network supported by identification of encoded transporter genes. Arrows indicate directionality of predicted metabolite exchange for amino acids (lysine, ornithine, glutamate, serine) and inorganic ions (Mg$^{2+}$, Mn$^{2+}$, Zn$^{2+}$) between species. Transporter genes shown on cell membranes (red = efflux, blue = uptake, purple = bidirectional) provide genomic support for predicted exchanges.

enzymes (GH66, GH87) and other glycoside hydrolases. These enzyme families are known to degrade extracellular polysaccharides and biofilm components in oral environments [53–57], though whether this function provides competitive advantages during colonization of the infant oral cavity remains to be experimentally validated.

Our genomic analyses of *Rothia* AIMSoral2 revealed enrichment of genes involved in fatty acid biosynthesis initiation compared to other infant-associated *Rothia* species. While the functional significance remains to be determined, fatty acid biosynthesis is associated with membrane synthesis and environmental adaptation in other bacteria [58, 59]. Genes encoding for GT11 cazymes (α(1,2/3)-fucosyltransferases) were also found to be enriched across *Rothia* AIMSoral2 genomes, which could contribute to host surface interactions based on their known functions in other bacterial systems [60, 61]. Together, these genomic features suggest *Streptococcus* AIMSoral1 and *Rothia* AIMSoral2 are adapted to the developing oral cavity, with distinct metabolic strategies providing ecological advantages during early colonization.

Metabolic modeling supports the potential for a cooperative relationship between *Streptococcus* AIMSoral1 and *Rothia* AIMSoral2. Within the three-species co-abundance module, metabolic complementarity patterns differed depending on the reference species: *Streptococcus* AIMSoral1 showed substantial complementarity with both partners, while *Rothia* AIMSoral2 and *Pauljensenia* sp900541895 each showed comparatively modest complementarity with *Streptococcus* AIMSoral1. This asymmetric pattern, combined with the predicted directionality of metabolite exchanges suggests that *Rothia* AIMSoral2 may primarily provide metabolites to *Streptococcus* AIMSoral1. While previous studies have shown beneficial interactions between these genera [17, 62, 63], interactions between these undescribed species in the infant oral cavity remained uninvestigated. Molecules predicted to be exchanged between *Streptococcus* AIMSoral1 and *Rothia* AIMSoral2 included nitrogen-rich amino acids (ornithine and lysine) and inorganic ions (Mg$^{2+}$, Mn$^{2+}$, Zn$^{2+}$) inferred to be transferred predominantly from *Rothia* AIMSoral2 to *Streptococcus* AIMSoral1. Genomic evidence supported these predictions, with all three species encoding the necessary membrane transporters. These metabolic exchanges may underpin their co-abundance in the infant oral microbiome, where *Rothia* AIMSoral2 provides nitrogen-rich amino acids and metal ions that support *Streptococcus* AIMSoral1 growth [64, 65] and enzymatic processes [66–68], while *Streptococcus* AIMSoral1 likely provides physical substrate through biofilm matrix formation that facilitates *Rothia* AIMSoral2 colonization [16, 69].

The identification of previously undescribed, prominent *Streptococcus* and *Rothia* species with specialized functional traits has significant implications for understanding early oral microbiome development and health outcomes. Both *Rothia* AIMSoral2 and *Pauljensenia* sp900541895 provide amino acids to *Streptococcus* AIMSoral1, including lysine, ornithine and glutamate, with the latter two being involved in arginine biosynthesis. Arginine production by *Streptococcus* AIMSoral1 may modulate oral pH, as arginine catabolism by other bacteria generates ammonia [70, 71], potentially providing pH buffering capacity that could help reduce caries risk [72, 73]. Similarly, *Rothia* AIMSoral2 possesses the ability to produce ammonia through nitrate reduction, suggesting its role in pH buffering and contributing to the maintenance of a balanced oral environment [5, 74]. Additionally, the exchange of glutamate and lysine within the network could influence pH regulation through polyamine (e.g., putrescine) metabolism, reinforcing metabolic interdependencies among these species. Consistent with a potential health-associated role, *Streptococcus* AIMSoral1 exhibited a negative co-abundance association with *Prevotella melaninogenica* (S8 Fig), a species implicated in periodontal disease in adults [33]. Beyond oral health,

nitrate metabolism by *Rothia* species is linked to NO production through the enterosalivary pathway, a process benefiting systemic cardiovascular health [5, 23].

Genome-resolved metagenomics is a powerful tool for identifying the functional elements of the microbiome and provide insights into potential ecological interactions of *Streptococcus* and *Rothia* species during early oral microbiome development. However, our functional interpretations are based on genomic predictions and homology to characterized enzymes from other organisms. Direct experimental validation through co-culture experiments and metabolomic profiling would be necessary to confirm these predicted metabolic interactions and cross-feeding relationships in the specific context of the infant oral microbiome. Additionally, MAGs represent incomplete genomes, potentially misrepresenting gene content. Most *Streptococcus* AIMSoral1 MAGs had medium-quality, which might have hindered functional annotations and downstream interpretations. Nevertheless, these MAGs clustered consistently with high-quality reference genomes in both gene presence/absence patterns and phylogenomic relationships, supporting our observations. Additionally, sample size potentially limited MAG recovery for lower-abundance species such as *Pauljensenia* sp900541895, precluding their functional characterization. Despite limitations, MAG reconstruction and metabolic modelling revealed the functional potential of undescribed oral species, generating testable hypotheses about species interactions for future infant oral microbiome studies.

Identifying novel species and their metabolic interactions has implications for understanding early-life oral microbiome assembly and health. Future research should prioritize the isolation of *Streptococcus* AIMSoral1 and *Rothia* AIMSoral2 and validate their predicted interactions, in particular regarding their pH homeostasis capabilities. Additionally, longitudinal studies that track these species from infancy through childhood will help clarify their long-term health impacts while revealing factors that influence their abundance dynamics. By deepening our knowledge of these ecological interactions, we ultimately can facilitate the development of precision-based interventions that promote both oral and systemic health.

## Materials and methods

### Ethics statement

The study was approved by the Medical Ethical Examination Committee of Amsterdam University Medical Center (METC Amsterdam UMC, Reference NL64399.018.17), with written informed consent obtained from all participants, and for infants, parental consent was secured after birth.

### Amsterdam infant microbiome study (AIMS) cohort

AIMS (https://aimsonderzoek.nl/en/) [75] is a microbiome-focused multi-ethnic, prospective birth cohort study by the GGD Amsterdam (Public Health Service of Amsterdam, the Netherlands) within the Sarphati Cohort [76].

### Study population and sample collection

We collected 117 non-invasive oral biosamples (tongue swab: n = 72, dental plaque: n = 45) for metagenomic shotgun sequencing from 24 AIMS mother-infant pairs (S1 Table). Mothers self-collected both their samples (at 34 weeks pregnancy) and their infants' (at 1 and 6 months postpartum). Tongue samples were collected by stroking a synthetic swab (FLO-QSwabs, Copan Italia s.p.a.) across the tongue dorsum for 30 seconds. Dental plaque was collected by using a sterile toothbrush (without toothpaste) to brush the buccal, lingual and occlusal surfaces of right lower molars for mothers and incisors in infants. Samples were home-stored in Liquid Amies Medium (eSwab) at -18/-20°C, transported at -4°C, and stored at -70°C. Questionnaires tested in a face validity study [77] gathered metadata on infant milk-feeding and weaning (S1 Table).

### Metagenomic sequencing and genome reconstruction

DNA was extracted using the ZY-D4306 ZymoBIOMICS-96 Magbead DNA Kit (with the use of the KingFisher Flex Purification System for high throughput) following BaseClear protocols. Of 117 samples, 84 yielded sufficient DNA for

sequencing: maternal (tongue = 22, dental plaque = 21), 1-month infants (tongue = 9, dental plaque = 0), and 6-month infants (tongue = 18, dental plaque = 14). Metagenomic libraries were sequenced on an Illumina NovaSeq platform, yielding 2x150bp paired-end reads.

Raw metagenomics reads were pre-processed using BBDuk and BBMap [78]. Raw reads were trimmed to remove adapters, poor quality bases (Q15) and reads shorter than 45 bp (minlength = 45). BBMap was used to discard human reads (minID = 0.95) and microbial reads were merged into longer single reads using BBMerge (min overlap = 16). The SPAdes [79] assembler (metagenomic mode) was used for metagenomic assembly.

For genome reconstruction, we used SemiBin for multi-sample binning to generate MAGs (SemiBin 1.0.0) [80]. For each mother-infant pair, we mapped short reads to a set of assembled contigs (500 bp) from the same mother-infant pair via Bowtie2 [81]. Samples were binned in multi-sample binning mode (*SemiBin multi_easy_bin*).

MAG quality was estimated using CheckM [81, 82] and GUNC v.0.1 [83] and genomes were taxonomically classified using the GTDB database. High-quality *Streptococcus* and *Rothia* MAGs were further passed through the *anvi-estimate-genome-completeness* module in Anvi'o v.8 [84] to estimate their completeness and redundancy; draft genomes were considered suitable for downstream analyses when displaying ≥50% completeness & ≤ 10% redundancy (i.e., medium, medium-high and high quality). The complete list of *Streptococcus* and *Rothia* MAGs used in this study are provided in S7 Table.

### Public reference genome acquisition

We conducted comparative genomic analyses on MAGs and publicly available *Streptococcus* and *Rothia* genomes from ProGenomes3 [85] and GTDB [86]. To ascertain the taxonomic placement of the undescribed *Streptococcus* species, we retrieved publicly available reference sequences from the *salivarius* group (*S. salivarius*: n = 5, *S. vestibularis*: n = 5, *S. thermophilus*: n = 5). For *Rothia*, we used type strains sequences or GTDB representatives (S7 Table). To perform comparative genomic analyses, we included *Streptococcus* and *Rothia* species detected in infant oral samples in this study and/or previously reported in infant oral microbiome literature, hereafter referred to as "infant-associated" [7, 8, 87–89].

After quality filtering, 110 *Streptococcus* (32 for *salivarius* group-specific analyses) and 63 *Rothia* MAGs and genomes (hereon referred to as genomes) were used for genetic relatedness analysis, while 57 *Streptococcus* and 53 *Rothia* genomes comprised the final dataset for pangenomics and metabolic enrichment analyses.

### Metagenomic species-profiling and strain tracking

In order to assess the relative abundance of bacterial taxonomic groups per sample, concatenated reads were mapped onto reference genomes from the GTDB R220 database [90] by using the computational tool sylph v.0.6.1 [91].

Single nucleotide variant (SNV) calling was used to determine overall strain-tracking between mothers and 1mo infants (i.e., vertical transmission) and mothers and 6mo infants (i.e., persisted vertical transmission or horizontal transmission). Reference-based strain tracking was performed using *inStrain* v1.0.0 [92] with reference genomes from the UHGG database, which we used to recruit quality-controlled merged reads using Bowtie2. Scaffold-level microdiversity metrics were calculated using the *inStrain profile* module. Pairwise species comparisons were performed using the *inStrain compare* module to calculate genome-level differences between samples (average nucleotide identity threshold = 0.999).

### Co-abundance network inference and correlation analyses

To identify potential species interactions within infant oral microbiomes, we performed co-abundance network inference using SPIEC-EASI (SParse InversE Covariance Estimation for Ecological Association Inference) via the iNAP 2.0 interface [93]. We analyzed species abundance profiles separately for 1-month tongue, 6-month tongue and 6-month tooth samples. Network inference was performed using Meinshausen-Bühlmann's neighborhood selection method with 20

penalty parameters only including species present in at least 5 samples. This threshold was determined through sensitivity analysis testing prevalence cutoffs from 2 to 6 samples; the 5-sample threshold was selected as it minimized discordance between SPIEC-EASI edges and significant Spearman correlations (S5 Fig and S3 Table). The minimal lambda was set to 0.01 for 1-month samples and 0.001 for 6-month samples, with a threshold of 0.05 for the StARS (Stability Approach to Regularization Selection) criterion for edge selection. Networks were visualized in Cytoscape, with edges representing significant positive associations and edge weights corresponding to covariance coefficients (S6 Fig). A subset network highlighting the focal species module (*Streptococcus* AIMSoral1, *Rothia* AIMSoral2, *Pauljensenia* sp900541895) and their positive and negative co-abundance associations was also generated (S8 Fig). To complement network analyses, we performed Spearman's rank correlation analyses on species with SPIEC-EASI interactions within 6-month tongue and 6-month tooth samples. One-month samples were excluded as no species pairs met the minimum prevalence threshold (≥5 samples). Correlation coefficients and p-values were calculated for all pairwise species comparisons, with significance determined at p ≤ 0.05. Only SPIEC-EASI co-abundance interactions for which Spearman's correlation yielded significant p-values were included in the final network. Spearman's correlation matrices and significance values are provided in Tables A and B in S5 Table as well as Tables A and B in S6 Table, while SPIEC-EASI outputs are in S4 Table.

## Phylogenomic and genetic relatedness analyses

Phylogenomic analyses were carried out in Anvi'o v8. Amino acid sequences of single-copy core genes (https://merenlab.org/2017/06/07/phylogenomics/) were extracted from *Streptococcus* and *Rothia* genomes (see Methods: Public reference genomes acquisition; S17 Fig), respectively, and aligned using the *anvi-get-sequences-for-hmm-hits* program. We further retrieved and aligned single-copy core genes from 31 genomes from the *salivarius* group for a targeted phylogenomic analysis of these *Streptococcus* species. We visualized the phylogenomics tree by using the resulting concatenated amino acid alignment as input for the *anvi-gen-phylogenomics-tree* program.

Genetic relatedness analyses were conducted for *Rothia* and the *salivarius* lineage within *Streptococcus* genus using the *anvi-compute-genome-similarity* module, which uses FastANI to calculate the average nucleotide identity (ANI). Genomes with an ANI ≥ 95% were classified as the same species.

To evaluate whether 16S rRNA-based approaches could reliably distinguish the novel species identified in this study, we extracted 16S rRNA gene sequences from reference genomes (*S. salivarius* (n = 3 genomes, 18 rRNA genes), *R. mucilaginosa* (n = 8 genomes, 10 rRNA genes), *R. mucilaginosa*_A (n = 2 genome, 3 rRNA genes), S. AIMSoral1 (n = 6 genomes, 8 rRNA genes) and R. AIMSoral2 (n = 2 genomes, 2 rRNA genes) and submitted them to the Silva aligner web tool (v1.2.11; https://www.arb-silva.de/aligner/) for taxonomic classification (S8 Table).

## *Streptococcus* and *Rothia* pangenome

Pangenome analyses were carried out in Anvi'o v8. Genomes were converted into contig databases (*anvi-gen-contig-database*) and functionally annotated from KEGG databases (*anvi-hmm* and *anvi-run-kegg-kofams*). Functionally annotated contig databases were stored (*anvi-gen-genomes-storage*) and used for pangenome reconstruction. Using the *anvi-pangenome* and *anvi-interactive* modules we reconstructed the *Streptococcus* and *Rothia* genus pangenome and subsequently visualize the hierarchical clustering of genomes based on gene presence/absence (Method: Ward; Distance: Euclidean; S11 and S13 Figs).

## Metabolic enrichment analyses

We analyzed high- and medium-quality genomes to identify metabolic differences among *Streptococcus* and *Rothia* species. Functional annotations were used to predict metabolic pathways (KEGG MODULE resource) for *Streptococcus* and *Rothia* species. For this purpose, we used the *anvi-estimate-metabolism* module, which predicts both presence and

completeness of the detected metabolic modules. Module abundance was determined as detailed in Watson et al. 2023 and the anvi'o tutorial https://anvio.org/m/anvi-estimate-metabolism [93]. Enrichment of metabolic modules across species was estimated using the *anvi-compute-metabolic-enrichment* module, resulting in enrichment scores and adjusted p-values (q-value) of modules found in *Streptococcus* and *Rothia* genomes [94].

### Profiling of carbohydrate-active enzymes and adhesion potential

High-quality infant-associated *Streptococcus* and *Rothia* genomes were annotated to the Carbohydrate-Active enZYmes (CAZy) database using Anvi'o v8 (run_dbCAN2) [95]. For adhesion potential, CAZy annotation identified carbohydrate-binding modules (CBMs), while lectin-encoding genes were determined using known *Streptococcus* adhesins from literature and LectomeXplore [96]. BLASTp (sequence identity≥40%, bitscore≥50, e-value≤0.001) [97] aligned genomes against lectin-sequences (S16 Table).

### Metabolic complementarity/interaction potential

To test whether species co-abundance was associated with metabolic interactions, we reconstructed phylogenetically-adjusted GEMS using PhyloMInt (v0.1.0) [98, 99]. Briefly, PhyloMInt employs CarveMe [28] to reconstruct GEMS and calculates (asymmetric) metabolic complementarity indices ($MI_{complementarity}$) for genomes pairs based on the metabolites required or produced by their metabolic modules. The $MI_{complementarity}$ quantifies the potential for metabolic cross-feeding between species pairs by measuring the proportion of essential metabolic compounds (seed compounds) required by one species that can be biosynthesized by another species' metabolic network. The metric ranges from 0 to 1, where higher values indicate greater theoretical potential for one species to support another through metabolite provisioning. A value of 1 represents complete biosynthetic complementarity (one species can provide all compounds that another requires but cannot produce itself), while 0 indicates no metabolic support potential. Importantly, $MI_{complementarity}$ is asymmetric: the complementarity of species A to B may differ from B to A, reflecting directional metabolic dependencies [98]. This metric provides an upper-bound estimate of cross-feeding potential based on genomic capacity and represents theoretical predictions that require experimental validation. We calculated the average $MI_{complementarity}$ for all species pairs tested differences using pairwise-wilcoxon tests.

To predict metabolic exchanges, we used the PhyloMInt PTM tool in iNAP 2.0 [99,100]. To validate the biological relevance of $MI_{complementarity}$ scores, we included a positive control using *Phocaeicola dorei* DSM17855 and *Lachnoclostridium symbiosum* WAL14673, species with experimentally confirmed metabolic interactions [41]. We obtained their genome assemblies and computed bidirectional $MI_{complementarity}$ scores using identical methodology (S12 Table). For metabolites predicted to be exchanged between *Streptococcus* AIMSoral1, *Rothia* AIMSoral2 and *Pauljensenia sp900541895*, we identified corresponding membrane transporters by querying KEGG annotations. Prior to transporter searches, we excluded coenzyme A derivatives and phosphorylated intermediates from the analysis, as these compounds are highly unlikely to be transported across membranes or exchanged between cells. For the remaining metabolites, we searched for relevant KEGG Orthology (KO) terms in the functional annotation matrices of these genomes (S14-S15 Tables and Fig 5D).

### Data analysis and statistics

Data analyses and statistical procedures were performed using a combination of RStudio (v2022.02.0+443) and Anvi'o (v8). Pangenomic hierarchical clustering, metabolic enrichment analyses using anvi-modules were performed in the Anvi'o 8.0 platform while other statistical tests were conducted in RStudio. False discovery rate ('*fdr*') correction was applied for multiple testing. For CAZyme, lectin/adhesin and enterosalivary pathway comparisons across species, differences in gene copy numbers (normalized by genome completeness) were tested using Kruskal-Wallis tests followed by pairwise Dunn's post-hoc tests. P-values were adjusted for multiple testing using *fdr* correction. Only high-quality genomes (≥90% completeness, ≤5% contamination) were included in statistical comparisons. Significance levels are indicated as: *p ≤ 0.05, **p ≤ 0.01, ***p ≤ 0.001, ****p ≤ 0.0001. Average $MI_{complementarity}$ indices were calculated for all species pairs and differences

were tested using pairwise Wilcoxon tests. All statistical outputs, including p-values, test statistics and *fdr*-corrected q-values, are provided in S3-S6 and S9-S11 Tables. A flowchart of the data processing pipeline from sample collection through downstream analyses is provided in S17 Fig. Metagenomics sample availability across all timepoints and sample types for each mother-infant pair is visualized in S18 Fig.

## Supporting information

**S1 Fig. Overview of the number of metagenomics reads retained per sample.** For each bar, colors indicate the proportion of reads from A) tongue and B) tooth biofilm samples that passed (green) and failed (red) quality control as well as reads mapping to human DNA (blue). Failed and human reads were subsequently filtered out the dataset.
(TIF)

**S2 Fig. Abundance and prevalence of bacterial species in the infant oral microbiome.** Mean relative abundance versus prevalence for bacterial species detected in infant oral samples. Labeled species represent the most abundant and prevalent taxa at each timepoint. (A) 1-month tongue (n = 9). (B) 6-month tongue (n = 18). (C) 6-month tooth (n = 14). *Streptococcus* AIMSoral1 and *Rothia* AIMSoral2 are among the most abundant and prevalent species at 6 months, along with other infant-associated streptococci and *Pauljensenia* sp900541895.
(TIF)

**S3 Fig. Principal coordinate analysis (PCoA) of tongue dorsum microbiomes from AIMS mothers (34wks gestation = brown) and their infants (1 month = green; 6 months = red).**
(TIF)

**S4 Fig. Oral microbiota of AIMS infants and corresponding milk feeding information.** A) PCoA of tongue dorsum (circles) and dental plaque (triangles) microbiomes from AIMS infants (1 month = small; 6 months = large). For each sample, the corresponding milk feeding type is reported (breastmilk = pink, mixed = red,formula = green). PERMANOVA analyses showed no significant clustering based on milk feeding within the 6 months samples (adonis: $R^2 = 0.07$, F = 1.10, p = 0.32). B) Relative abundances of *Streptococcus* AIMSoral1, *Rothia* AIMSoral2 and *Pauljensenia* sp900541895 found in infants breast-, formula- and mixed milk-feeding. Significance among groups was tested for each species by pairwise Wilcoxon's signed rank tests. No significant differences were detected. C) Correlation plot of *Streptococcus* AIMSoral1 and *Rothia* AIMSoral2 showed no association between their abundance and milk-feeding type.
(TIF)

**S5 Fig. Sensitivity analysis for determining species prevalence threshold in co-abundance network inference.** Co-abundance networks were constructed using SPIEC-EASI for species present above different prevalence thresholds (>1 to >5 samples) and corroborated using Spearman's rank correlation. Bars show the number of positive network interactions (edges/links) identified by SPIEC-EASI at each threshold. Blue segments represent edges supported by significant Spearman correlations ($p \le 0.05$; "kept"), while orange segments represent edges lacking Spearman significance ("removed"). The red dashed line indicates the selected threshold (>4 samples), chosen to minimize discordance between SPIEC-EASI edges and Spearman correlation significance while maintaining network complexity. A) 1-month tongue samples (n = 9). B) 6-month tongue samples (n = 18). C) 6-month tooth samples (n = 14). At 1 month, no significant co-abundance relationships were detected regardless of threshold.
(TIF)

**S6 Fig. Co-abundance interaction network of infant tongue bacterial species highlighting focal undescribed species and their interacting partners.** SPIEC-EASI network from 6-month infant tongue samples. Network inference was carried out only on species present in at least 5 samples. Only network links supported by significant Spearman's rank correlations are shown. Orange nodes: undescribed *Streptococcus* AIMSoral1, *Rothia* AIMSoral2 and co-occurring

*Pauljensenia sp900541895* selected for metabolic modeling. Blue nodes: other bacterial species. Edges width indicates the strength of the association based on SPIEC-EASI covariance.
(TIF)

**S7 Fig. Focal species and their positive and negative co-abundance interactions in the 6-month infant tongue network.** Subset of the SPIEC-EASI network from 6-month infant tongue samples showing direct interactions of focal species. Orange nodes: undescribed *Streptococcus* AIMSoral1, *Rothia* AIMSoral2 and co-occurring *Pauljensenia* sp900541895 selected for metabolic modeling. Blue nodes: species exhibiting negative co-abundance associations with focal species. Green edges indicate positive associations; red edges indicate negative associations. Edge labels show SPIEC-EASI covariance values.
(TIF)

**S8 Fig. Heatmaps showing Spearman's rank correlation coefficients for infant oral species at 6 months of age.** Correlations between oral species in A) the tongue biofilm and B) dental biofilm of 6-month-old AIMS infants are colored in blue (positive) or red (negative). Significance is indicated by asterisks: *p ≤ 0.05, ** ≤ 0.01, *** ≤ 0.001, **** ≤ 0.0001. Spearman's rank correlations were carried out only among species present in at least 5 samples per group.
(TIF)

**S9 Fig. Phylogenomic tree of the *Streptococcus* genus.** Phylogenomic analysis of 110 *Streptococcus* genomes representing 89 species, including 89 reference genomes and 21 metagenome-assembled genomes (MAGs). *Lactococcus lactis* was included as an outgroup. Red circles indicate bootstrap values ≥ 0.9.
(TIF)

**S10 Fig. Heatmaps of Average Nucleotide Identity (ANI) of *Streptococcus* (*salivarius* group) and *Rothia* genomes.** Pairwise ANI comparisons between A) *Streptococcus* (*salivarius* group) and B) *Rothia* MAGs (white) and reference genomes (red). For each genome, the GTDB/AIMSoral and final species assignments are provided.
(TIF)

**S11 Fig. Pangenome of infant-associated *Streptococcus* species in the oral cavity of AIMS infants.** The figure shows the pangenomic characteristics of infant-associated *Streptococcus* species sorted by gene presence/absence. The pangenome includes MAGs reconstructed from AIMS oral samples (sampling location: tongue dorsum = purple, tooth plaque = orange) collected at different timepoints (infant 1 month = green, infant 6 months = red). No high- and medium-quality MAGs from infant-associated species were retrieved from maternal oral samples. Reference genomes are given in grey. Species annotations as well as bars showing genome/MAG total length, completion and redundancy are shown for each strain. For each gene cluster, metrics such as number of genomes where a gene cluster is present ('num of contributing genomes) and various functional annotations are shown.
(TIF)

**S12 Fig. Amino acid biosynthesis modules distinguish *Streptococcus* AIMSoral1 and *S. salivarius* from other infant-associated streptococci.** Presence/absence heatmap of 15 amino acid biosynthesis modules (KEGG) across 57 infant-associated *Streptococcus* genomes from six species. Red indicates module presence; white indicates absence. Hierarchical clustering (top dendrogram) groups genomes by metabolic similarity, with species identity and genome completeness shown by color. *Streptococcus* AIMSoral1 and *S. salivarius* uniquely retain arginine, histidine, serine, and ornithine biosynthesis pathways that are absent in *S. mitis*, *S. oralis*, *S. peroris* and *S. lactarius*.
(TIF)

**S13 Fig. Pangenome of infant-associated *Rothia* species in the oral cavity of AIMS infants.** The figure shows the pangenomic characteristics of infant-associated *Rothia* species sorted by gene presence/absence. The pangenome

includes MAGs reconstructed from AIMS oral samples (sampling location: tongue dorsum = purple, tooth plaque = orange) collected at different timepoints (mother 34wks gestation = brown, infant 1 month = green, infant 6 months = red). Reference genomes are given in grey. Species annotations as well as bars showing genome/MAG total length, completion and redundancy are shown for each strain. For each gene cluster, metrics such as number of genomes where a gene cluster is present ('num of contributing genomes) and various functional annotations are shown.
(TIF)

**S14 Fig. Metabolic complementarity and interactions among infant-associated *Streptococcus* and *Rothia* species in our study.** A) Bargraphs show the average metabolic complementarity index (±SD) for pairwise interactions among infant-associated *Streptococcus* and *Rothia*, and vice versa. For each complementarity species pair, the reference species is shown on the x-axis, while the other species is shown in the facet on top of the bar. B) Heatmap showing bacterial-metabolite bipartite interactions production = red, utilization = blue, both=purple, no interaction = white) potentially occurring among *Streptococcus* AIMSoral1, *Rothia* AIMSoral2 and their closely related infant-associated species in our study. Metabolites are classified by their metabolite type.
(TIF)

**S15 Fig. Metabolic interactions among *Streptococcus* AIMSoral1, *Rothia* AIMSoral2 and *Pauljensenia* sp900541895.** Heatmap showing unfiltered PhyloMInt PTM output of bacterial-metabolite bipartite interactions (production = red, utilization = blue, both=purple, no interaction = white) potentially occurring among species in the tongue network. Metabolites are classified by their metabolite type. Also intracellular CoA and phosphate derivatives, unlikely to be transferred, are included in the heatmap.
(TIF)

**S16 Fig. Abundance of *Streptococcus* AIMSoral1 and *Rothia* AIMSoral2 across infant oral samples from Ferretti et al. 2018.**
(TIF)

**S17 Fig. Step-by-step workflow of metagenomic data processing and analysis.** Comprehensive flowchart showing the analytical pipeline from sample collection through final analyses. The workflow is organized into six color-coded stages: sample preparation, sequencing and genome reconstruction, abundance-based analyses, genomic analyses, functional analyses and metabolic modeling. Beginning with 117 oral biosamples from 24 mother-infant pairs, the pipeline includes metagenomic sequencing, MAG reconstruction and quality control, followed by taxonomic profiling, co-abundance analysis, strain tracking, comparative genomics, functional characterization and metabolic modeling. Sample sizes for each analytical step are detailed in each box and at the bottom of the figure.
(TIF)

**S18 Fig. Overview of metagenomics sample availability across timepoints and sample types.** Heatmap showing sample availability for the 24 mother-infant pairs enrolled in the AIMS cohort. Rows represent individual participants (P1-P24) and columns represent sample collection timepoints and oral sites. Green: sample available for sequencing; white: sample not available. Cyan: tongue swab; pink: tooth plaque. Maternal samples were collected during pregnancy (third trimester), and infant samples were collected at 1 month (tongue only; no erupted teeth) and 6 months (tongue and tooth when dentition present).
(TIF)

**S1 Table. Metadata for metagenomics samples retrieved from AIMS mothers (34wks gestation) and their infants (1 and 6 months) available for this study.**
(XLSX)

**S2 Table. Comprehensive strain-level tracking analysis of oral bacteria using inStrain.** Summary of inStrain strain tracking output showing bacterial strain sharing patterns across multiple comparison types: (1) within-individual sample-to-sample sharing (tongue vs. plaque), (2) within-individual temporal stability (1-month and 6-month time points), and (3) vertical transmission between mothers and infants at 1 month and 6 months postpartum. Results indicate whether identical bacterial strains were detected between sample pairs.
(XLSX)

**S3 Table. Pairwise species co-abundance interactions in the 6-month infant tongue microbiome identified by SPIEC-EASI and validated by Spearman's rank correlation.** Species pairs showing significant co-abundance in SPIEC-EASI network analysis that were also supported by significant Spearman's rank correlation (p ≤ 0.05). Interactions are ranked by inverse covariance coefficient (iCov) in descending order, where higher absolute values indicate stronger associations. node1 and node2: species pairs; iCov: inverse covariance coefficient from SPIEC-EASI; p-value: p-value from Spearman's rank correlation.
(XLSX)

**S4 Table. SPIEC-EASI sparse inverse covariance matrices underlying bacterial interaction networks in the oral microbiomes of 6-months-old AIMS infants (tongue dorsum and dental plaque, separately).**
(XLSX)

**S5 Table. Spearman rank-order correlation coefficients (Rho) for species pairs in co-abundance network analysis. Spearman's rank correlation coefficients (Rho) for pairwise abundance correlations between species included in SPIEC-EASI network inference from 6-month infant oral samples.** A) Tongue dorsum samples (n = 18), comprising 36 species present in at least 5 samples and B) dental plaque samples (n = 14) comprising 22 species present in at least 5 samples.
(XLSX)

**S6 Table. Spearman rank-order correlation p-values for species pairs in co-abundance network analysis. Spearman's rank correlation p-values for pairwise abundance correlations between species included in SPIEC-EASI network inference from 6-month infant oral samples.** A) Tongue dorsum samples (n = 18), comprising 36 species present in at least 5 samples and B) dental plaque samples (n = 14) comprising 22 species present in at least 5 samples.
(XLSX)

**S7 Table. Overview of medium- and high-quality metagenome-assembled genomes (MAGs) and genomes used for this study.**
(XLSX)

**S8 Table. Taxonomic annotation performance of Silva aligner for 16S rRNA sequences from Streptococcus and Rothia genomes.** Validation of Silva aligner (v1.2.11) taxonomic classification accuracy using 16S rRNA gene sequences extracted from genomes/MAGs used in this study. 16S rRNA sequences were extracted from *S. salivarius* (n = 3 genomes, 18 rRNA genes), *R. mucilaginosa* (n = 8 genomes, 10 rRNA genes), *R. mucilaginosa*_A (n = 2 genome, 3 rRNA genes), *Streptococcus* AIMSoral1 (n = 6 genomes, 8 rRNA genes) and *Rothia* AIMSoral2 (n = 2 genomes, 2 rRNA genes). Columns show: sequence_identifier (rRNA gene location in genome), genome annotation (GTDB-based species annotation), identity (% identity to Silva reference) and taxonomic assignments from five databases (lca_tax_embl_ebi_ena, lca_tax_gtdb, lca_tax_ltp, lca_tax_rdp, lca_tax_slv).
(XLSX)

**S9 Table. Pairwise Dunn's test results for gene copy number differences among infant oral *Streptococcus* and *Rothia* species.** Statistical comparison of gene copy numbers across infant-associated *Streptococcus* (six species) and

*Rothia* (three species) genomes using Dunn's post-hoc test. Comparisons include genes involved in nitrate reduction (NO3-NO2-NO pathway), carbohydrate metabolism (CAZymes), adhesion and broader metabolic module pathways. (XLSX)

**S10 Table. Output of the anvi'o metabolic enrichment analysis performed among infant-associated *Streptococcus* species.**
(XLSX)

**S11 Table. Output of the anvi'o metabolic enrichment analysis performed among infant-associated *Rothia* species.**
(XLSX)

**S12 Table. PhyloMInt complementarity indices for pairwise comparisons between infant-associated *Streptococcus*, *Rothia* genomes, *Pauljensenia sp900541895* and positive controls.**
(XLSX)

**S13 Table. Metabolite production and utilization profiles for oral bacterial community members.** (A) Node attributes for a network comprising five oral bacterial species (*Streptococcus salivarius*, *Streptococcus* AIMSoral1, *Rothia mucilaginosa*, *R. mucilaginosa*_A, and *Rothia* AIMSoral2) and 63 associated metabolites. (B) Node attributes for a network comprising three oral bacterial species (*Pauljensenia* sp900541895, *Streptococcus* AIMSoral1 and *Rothia* AIMSoral2) and 44 associated metabolites. Both tables include node type classifications (microbe or metabolite), BiGG identifiers, metabolite superclass annotations, associated BiGG model references and external database cross-references.
(XLSX)

**S14 Table. Copy numbers of membrane transporter enzymes involved in metabolite exchange among oral bacteria.** The table shows KEGG Ortholog (KO) copy numbers for enzymes predicted to facilitate uptake and efflux of exchanged metabolites in three oral bacterial species: *Streptococcus* AIMSoral1 (independently assembled MAG VTCC12814), *Rothia* AIMSoral2 (MAG_i6P11_Tongue) and *Pauljensenia sp900541895* (independently assembled MAG UMGS402).
(XLSX)

**S15 Table. Overview of predicted metabolite interactions among *Streptococcus* AIMSoral1 (independently assembled MAG VTCC12814), *Rothia* AIMSoral2 (MAG_i6P11_Tongue), and *Pauljensenia sp900541895* (independently assembled MAG UMGS402) validated through genomic identification of transporter genes.** For each species-metabolite pair, the table presents: (i) the predicted metabolite attribute (secrete/utilization) from metabolic modeling; (ii) identified transporter genes (KEGG Orthology identifiers); (iii) transporter functional annotations; (iv) transporter directionality (import/export/bidirectional) determined from literature; and (v) validation status indicating whether the transporter direction matches the predicted metabolite attribute (MATCH: concordance; MISMATCH: discordance; NA: no transporter identified).
(XLSX)

**S16 Table. Overview reference adhesin/lectins protein sequences used for the genomic mining of adhesins across infant-associated *Streptococcus* spp.**
(XLSX)

**S17 Table. Matrices of KEGG Orthologs (KOs) copy numbers per infant-associated *Streptococcus* and *Rothia* genomes/MAGs.**
(XLSX)

**S18 Table. AIMS oral microbiome data.** Matrix containing the relative abundances of bacteria species detected in AIMS oral samples (tongue dorsum and dental plaque, when applicable) from mothers (34wks gestation) and their infants at 1 and 6 months of age.
(XLSX)

## Acknowledgments

We thank all the families who participate in the Amsterdam Infant Microbiome Study (AIMS) cohort and the entire research team responsible for the recruitment of the cohort and collection of samples and questionnaires.

## Author contributions

**Conceptualization:** Nicholas Pucci, Egija Zaura, Daniel R. Mende.

**Data curation:** Nicholas Pucci, Amke Marije Kaan.

**Formal analysis:** Nicholas Pucci, Daniel R. Mende.

**Funding acquisition:** Arnoud P. Verhoeff, Egija Zaura, Daniel R Mende.

**Investigation:** Nicholas Pucci, Amke Marije Kaan, Daniel R Mende.

**Methodology:** Nicholas Pucci, Daniel R. Mende.

**Resources:** Joanne Ujčič-Voortman, Arnoud P. Verhoeff.

**Software:** Nicholas Pucci.

**Supervision:** Egija Zaura, Daniel R. Mende.

**Validation:** Amke Marije Kaan, Daniel R. Mende.

**Visualization:** Nicholas Pucci.

**Writing – original draft:** Nicholas Pucci, Daniel R. Mende.

**Writing – review & editing:** Nicholas Pucci, Arnoud P. Verhoeff, Egija Zaura, Daniel R. Mende.

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
