## [Decision Letter · Decision Letter 0]

26 Sep 2025

Unique ecology of co-occurring functionally and phylogenetically undescribed species in the infant oral microbiome

PLOS Computational Biology

Dear Dr. Mende,

Thank you for submitting your manuscript to PLOS Computational Biology. After careful consideration, we feel that it has merit but does not fully meet PLOS Computational Biology's publication criteria as it currently stands. Therefore, we invite you to submit a revised version of the manuscript that addresses the points raised during the review process.

Please submit your revised manuscript within 90 days Nov 26 2025 11:59PM. If you will need more time than this to complete your revisions, please reply to this message or contact the journal office at ploscompbiol@plos.org. Please include the following items when submitting your revised manuscript:

We look forward to receiving your revised manuscript.

Kind regards,

Boyang Ji, Ph.D.

Academic Editor

PLOS Computational Biology

Ilya Ioshikhes

Section Editor

PLOS Computational Biology

**Journal Requirements:**

At this stage, the following Authors/Authors require contributions: Amke Marije Kaan, Joanne Ujčič-Voortman, Arnoud P. Verhoeff, Egija Zaura, and Daniel Mende. Please ensure that the full contributions of each author are acknowledged in the "Add/Edit/Remove Authors" section of our submission form.

- ® on page: 17 and 18.

4) Please ensure that the funders and grant numbers match between the Financial Disclosure field and the Funding Information tab in your submission form. Note that the funders must be provided in the same order in both places as well.

**Reviewers' comments:**

Reviewer's Responses to Questions

**Comments to the Authors:**

Reviewer #1: Below are my comments and I have uploaded it as a separate document as well.

1. It would strengthen the manuscript to expand the Introduction by clarifying the purpose/rationale and relevance of the network analysis. Currently, the rationale for incorporating co-occurrence networks and how they contribute to the interpretation of microbial interactions is somewhat lacking. Providing additional information would help justify this analytical approach and enhance the overall objective of the study.

2. The manuscript presents several layers of data pre-processing and quality control steps, but these are currently spread across many short subsections, which may reduce readability. For instance, sections from “Pre-processing of metagenomic reads” to “Metagenomic species profiling and strain tracking” could be considered as part of a unified pre-processing workflow. To improve clarity, I suggest consolidating these into a single subsection or grouping them under a “Pre-processing” section. Additionally, I strongly recommend including a flowchart or schema figure summarizing the step-by-step data processing pipeline. This would greatly enhance the transparency and reproducibility of the study, as is commonly done in bioinformatics studies.

3. Since the network is inferred based on abundance or taxa count data, it would be more accurate to describe it as a co-abundance network rather than a co-occurrence network. I suggest clarifying this point in the Methods section and also reflecting this terminology when presenting the rationale for the network analysis in Introduction section

4. The current network analysis is based on Spearman’s rank correlations, which represent marginal (unadjusted) pairwise associations and do not account for the presence of other taxa when estimating co-abundance relationships. This is analogous to interpreting univariable regression results in the presence of multiple covariates. Given the compositional and interdependent nature of microbial communities, I strongly encourage the authors to consider whether partial correlation-based approaches or more advanced network inference methods would be more appropriate. Established alternatives such as SPIEC-EASI (graphical model-based) or regression-based methods like SOHPIE can better account for indirect effects and improve the specificity of inferred interactions.

5. As noted earlier, co-occurrence or species interaction patterns cannot be fully captured using marginal (unadjusted) correlation-based methods. The authors state that “pairwise co-occurrence patterns across ages” were inferred using hierarchical clustering and Spearman’s rank correlations. However, this approach does not account for the compositional nature of microbiome data, nor does it adjust for potential confounders such as age or interactions with other taxa. As a result, the strength of connections (edges) in the resulting network may reflect spurious associations driven by shared abundance patterns or unmeasured covariates. I encourage the authors to consider network inference methods that can incorporate covariate adjustment or control for other taxa, to more robustly infer species interactions.

6. The manuscript states that no metabolic module was exclusively enriched in Streptococcus AIMSoral1, yet a detailed discussion follows regarding its distinct amino acid metabolic profile. While this section is informative, the interpretation may come across as stronger than what the data directly support. It would be helpful either to reframe this discussion more cautiously or to include differential abundance analyses to substantiate the claim.

7. The manuscript uses MIcomplementarity to infer species interactions, but it is unclear how this metric should be interpreted in a biological or statistical sense. It would be helpful to clarify what the scores represent, whether they have a defined scale or units, and how they were validated. In particular, do these scores correspond to any known or experimentally supported interspecies metabolic interactions?

8. The identification of CAZymes such as GH66 and GT11 adds genomic depth, but the downstream functional interpretations like mucosal colonization or EPS degradation seem speculative in the absence of expression data or functional assays. It may strengthen the manuscript to either soften these interpretations or clarify the limitations if you agree.

9. Most results are presented at the species level, which is a common approach in metagenomic studies. However, given that genome-resolved analysis allows for finer resolution, it may be worth clarifying whether strain-level differences were considered or whether species-level resolution was sufficient for the study’s objectives.

10. The manuscript includes metabolic enrichment analysis as a downstream step, but its role within the overall study framework is not clearly articulated. A brief explanation of the rationale for including this analysis and how it relates to or complements the co-abundance network approach would help clarify what additional biological insights it offers beyond the network analysis.

11. In the Results section, several additional analyses are presented, including phylogenomic, pangenome, and enrichment analyses. These appear to serve as supplementary layers to the main objective of the study, which focuses on co-abundance network analysis. For clarity and improved readability, it may be helpful to group these components into a dedicated subsection or a separate section titled something like “Additional Downstream Analyses.”

12. In this study, authors primarily focused on the most prevalent species. Have the authors examined whether similar or substantially different patterns emerge when considering less abundant species that are not overly sparse and have been highlighted in recent clinical studies for their biological relevance? Including this as a sensitivity analysis could help assess the robustness of the findings and offer additional insights into underrepresented taxa.

Reviewer #2: In this work, authors performed a metagenomic analysis on infant and mother oral microbiomes.

1. In general, all analysis processed were implemented on existing tools and methods, with limited novelty on methodology, which can hardly fit the scope of PLoS CB journal.

2. The study highlights a shift in microbial composition from 1 to 6 months, but the drivers of this shift (e.g., dentition, dietary transitions, maternal transmission) remain underexplored. Longitudinal analyses with more frequent sampling points could clarify how developmental milestones influence the emergence of the novel species.

3. Furthermore, while strain sharing between mothers and infants is noted as rare, investigating vertical transmission of functional traits (rather than just strains) may reveal hidden maternal influences

4. While this is a computational study, the paper would benefit from acknowledging the lack of experimental validation (e.g., co-culture, metabolomics) and explicitly outlining which predictions could be tested in future studies.

5. The metabolic modeling approach (GEMS + PhyloMInt) supports potential nutrient exchanges between co-occurring species. However, the conclusions at times read as causal. Please rephrase key statements to clarify that these are predictions rather than confirmed interactions.

6. The link of source code and source data is not available.

Reviewer #3: In this article, Pucci et al., report novel co-occurring species and provide metapangenomics and GEMS based evidence to suggest functional interactions – of relevance to our understanding of early oral microbiome development.

This work represents a valuable addition to our understanding of human oral microbiome, particularly of its development during early infancy. The study has many strengths including its longitudinal nature, access to very valuable mother-infant dyad and tongue dorsum and dental plaque samples, as well as metagenomic sequencing – enabling the identification of reported “novel” species and their potential functional interactions. However, in my view, the manuscript does not provide sufficient details to qualify all the reported observations.

Specifically, to begin with- how many samples were infact studied longitudinally? They report 24 mother-infant dyad but seemingly only 9 were available at 1-month and 14 at 6-months? Numbers were unclear for tongue versus dental as well. It would help to have a clear figure describing the sample sizes available/used per analyses to inform the reader. This is important for statistical considerations, which were absolutely missing from all figures/ legends/ methods section of the manuscript.

How were the relative abundances/ prevalences defined? Lines 188-189: authors say they focussed on the functional characterization of Strept oral 1 and Rothia oral2 but not Veionella Oral 3 and Pauljensenia Oral 4 because of high abundance and prevalence but how was that determined? And where are these data? Is it possible that lower sample sizes may have affected abundances/prevalences?

Similarly, functional enrichments were hard to follow statistically. Was each band significantly different (e.g., Fig 3 and 4). It seemed that most species shown in Fig 3 had the potential for amino acid metabolism and fatty acid biosynthetic potential was not exclusive to Rothia oral 2 as part of Fig 4. Similarly, in Figure 5, the metabolite production versus utilization was not specific to Rothia Oral 2 and Strept Oral 1 but was essentially shared among the 4 species of which the other 2 were not studied functionally. The conclusions drawn are limited to the analyses undertaken, however, there is potential for further data mining or adding stronger rationale for the current conclusions.

The manuscript will also benefit from further details in most figures and legends, with better referencing to the figures and their respective legends. The reported observations will also benefit from adding analyses at 9 month or 12 months of age, if at all those data are available. Or else from a description of why that may or may not be the case. Additionally, the limitations of the current work must be considered.

**Have the authors made all data and (if applicable) computational code underlying the findings in their manuscript fully available?**

Reviewer #1: **No:** Full access to the data requires a data use agreement. The authors have noted the source and provided the hyperlink to the database.

Reviewer #2: **No:**

PLOS authors have the option to publish the peer review history of their article (what does this mean? ). If published, this will include your full peer review and any attached files.

**Do you want your identity to be public for this peer review?** For information about this choice, including consent withdrawal, please see our Privacy Policy .

Reviewer #2: No

Reviewer #3: No

**Figure resubmission:**

**Reproducibility:**



---

## [Decision Letter · Decision Letter 1]

18 Feb 2026

Dear Dr Mende,

We are pleased to inform you that your manuscript 'Unique ecology of co-occurring functionally and phylogenetically undescribed species in the infant oral microbiome' has been provisionally accepted for publication in PLOS Computational Biology.

Best regards,

Boyang Ji, Ph.D.

Academic Editor

PLOS Computational Biology

Ilya Ioshikhes

Section Editor

PLOS Computational Biology

Reviewer's Responses to Questions

**Comments to the Authors:**

Reviewer #1: I would like to thank the authors for their thorough and thoughtful revision of the manuscript. Authors have addressed my previous comments and concerns very carefully, providing detailed explanations, clear rationales where there were points of disagreement, revised figures in response to my suggestions, a strengthened Discussion section, and substantial improvement and clarification of the network analysis component, including the incorporation of SPIEC-EASI.

I have no further comments or concerns at this time and very much enjoyed reading your work.

Reviewer #2: All my previous comments have been addressed.

**Have the authors made all data and (if applicable) computational code underlying the findings in their manuscript fully available?**

Reviewer #1: Yes

Reviewer #2: Yes

PLOS authors have the option to publish the peer review history of their article (what does this mean? ). If published, this will include your full peer review and any attached files.

**Do you want your identity to be public for this peer review?** For information about this choice, including consent withdrawal, please see our Privacy Policy .

Reviewer #1: No

Reviewer #2: No

---

## [Editor Report · Acceptance letter]

PCOMPBIOL-D-25-01114R1

Unique ecology of co-occurring functionally and phylogenetically undescribed species in the infant oral microbiome

Dear Dr Mende,

I am pleased to inform you that your manuscript has been formally accepted for publication in PLOS Computational Biology. Your manuscript is now with our production department and you will be notified of the publication date in due course.

With kind regards,

Anita Estes
